# Observation of the topological soliton state in the Su–Schrieffer–Heeger model

Eric J. Meier[1], Fangzhao Alex An[1] & Bryce Gadway[1]

The Su–Schrieffer–Heeger (SSH) model, which captures the most striking transport properties of the conductive organic polymer *trans*-polyacetylene, provides perhaps the most basic model system supporting topological excitations. The alternating bond pattern of polyacetylene chains is captured by the bipartite sublattice structure of the SSH model, emblematic of one-dimensional chiral symmetric topological insulators. This structure supports two distinct nontrivial topological phases, which, when interfaced with one another or with a topologically trivial phase, give rise to topologically protected, dispersionless boundary states. Here, using $^{87}$Rb atoms in a momentum-space lattice, we realize fully tunable condensed matter Hamiltonians, allowing us to probe the dynamics and equilibrium properties of the SSH model. We report on the experimental quantum simulation of this model and observation of the localized topological soliton state through quench dynamics, phase-sensitive injection, and adiabatic preparation.

[1] Department of Physics, University of Illinois at Urbana-Champaign, Urbana, Illinois 61801-3080, USA. Correspondence and requests for materials should be addressed to B.G. (email: bgadway@illinois.edu).

Remarkably, the conductivity of the polymer *trans*-poly-acetylene can be increased by over 10 orders of magnitude through halogen doping, transforming it from a simple organic insulator into a metallic conductor[1–3]. This unusual electronic property stems from topologically protected solitonic defects that are free to move along the polymer chain[4,5]. To account for such behaviour, Su, Schrieffer and Heeger (SSH) proposed a simple one-dimensional (1D) tight-binding model with alternating off-diagonal tunnelling strengths to represent doped polyacetylene[4]. The SSH model has since served as a paradigmatic example of a 1D system supporting charge fractionalization[5,6] and topological character[7].

The emergence of such exotic phenomena in a simple 1D setting has naturally inspired numerous related experimental investigations, including efforts to probe aspects of the SSH model by quantum simulation with pristine and tunable ultracold atomic gases. Recently, using real-space superlattices[8,9], several bulk characteristics of the SSH model's topological nature have been explored. These include the measurement of bulk topological indices[10] and the observation of topologically robust charge pumping[11,12]. Topological pumping has also been observed in a 'magnetic lattice' based on internal-state synthetic dimensions[13,14]. Cold atom experiments have even begun to probe topological boundary states, with recent evidence for boundary localization in the related 1D Dirac Hamiltonian with spatially varying effective mass[15]. Highly tunable photonic simulators[16] have provided a complementary window into the physics of topological systems[17,18]. In particular, evidence for topological 1D bound states has been found[19] in the discrete quantum walk of light in a Floquet-engineered[20] system resembling the SSH model. Related bound state behaviour has also been observed in 1D photonic quasicrystals[21,22].

Here, using an atom-optics[23,24] realization of lattice tight-binding models[25,26], we report on the direct imaging and probing of topological bound states in the SSH model through quench dynamics, phase-sensitive injection, and adiabatic preparation. Our technique, based on the controlled coupling of discrete atomic momentum states through stimulated Bragg transitions, allows for arbitrary and dynamic control over the tunnelling amplitudes, tunnelling phases, and on-site energies in an effective 1D tight-binding model[25,26]. We use these unique capabilities to prepare, probe, and directly image topological boundary states of the SSH model with unprecedented resolution and control.

The molecule *trans*-polyacetylene consists of a 1D carbon chain connected through alternating single and double bonds. This sublattice bond structure, emblematic of 1D chiral symmetric topological insulators[7], leads to two distinct topological phases. Interesting electronic properties arise when these two phases (or one of the phases and a trivial, nontopological phase) are interfaced. Figure 1a shows an example of an edge defect carbon atom interfacing a polyacetylene chain with the nontopological vacuum. In the case of a central defect, the two distinct topological phases of the SSH model are interfaced at a defect carbon atom with two single bonds, as illustrated in Fig. 1b.

Topological polyacetylene chains support zero-energy electronic eigenstates localized to such defects, the basis of which may be found by examining the effective 1D tight-binding model proposed by SSH in ref. 4. In this simplified picture (see Fig. 1a,b), the polymer chain's carbon backbone acts as a 1D lattice for electrons, with the alternating double and single bonds represented as strong $(t + \Delta)$ and weak $(t - \Delta)$ tunnelling links, respectively. The Hamiltonian describing the SSH model is given by

$$H = -(t + \Delta) \sum_{n \in \text{odd}} \left( c_{n+1}^{\dagger} c_n + \text{h.c.} \right) \\ -(t - \Delta) \sum_{n \in \text{even}} \left( c_{n+1}^{\dagger} c_n + \text{h.c.} \right), \quad (1)$$

where $t$ is the average tunnelling strength and $2\Delta$ the tunnelling imbalance. This bipartite sublattice structure, the result of a Peierls distortion in the polyacetylene chain, leads to a two-band energy dispersion as in Fig. 1(c, inset) with an energy gap $E_{\text{gap}} = 2\Delta$. When distinct topological phases of these SSH wires are directly interfaced, spatially localized 'mid-gap' eigenstates appear in the middle of this energy gap.

Figure 1c displays such a localized mid-gap state wavefunction for the case of an edge (site zero) defect as in Fig. 1a, with the particular choice of $\Delta/t = 0.41$. Several key features of the topological boundary state are illustrated by Fig. 1c. First, it is localized to the defect site with a characteristic decay length $\xi/d \sim (\Delta/t)^{-1}$ due to its energetic separation by $\Delta$ from the dispersive bulk states, where $d$ is the spacing between lattice sites. Additionally, this topological boundary state exhibits the absence of population on odd lattice sites and a sign inversion of the wavefunction on alternating even sites. Both of these features can be understood from the fact that the state is composed of two quasimomentum states with $q = \pm \pi/2d$, leading to a $\cos(\pi n/2d)$-like variation of the eigenstate wavefunction underneath the aforementioned exponentially decaying envelope. Below, we directly explore these properties of the mid-gap state wavefunction through single-site injection, multi-site injection, and adiabatic preparation.

## Results

**Overview.** We begin with a brief description of our experimental methods for studying the SSH model, as discussed previously in refs 25,26. We initiate momentum-space dynamics of [87]Rb condensate atoms through controlled, time-dependent driving with an optical lattice potential formed by lasers of wavelength $\lambda = 1,064$ nm and wave number $k = 2\pi/\lambda$. The lasers coherently couple 21 discrete atomic momentum states, creating a 'momentum-space lattice' of states in which atomic population may reside. The momentum states are characterized by site indices $n$ and momenta $p_n = 2n\hbar k$ (relative to the lowest momentum value). The coupling between these states is fully controlled through 20 distinct two-photon Bragg diffraction processes, allowing us to simulate tight-binding models with arbitrary, local, and time-dependent control of all site energies and tunnelling terms[25,26]. This control enables the creation of hard-wall boundaries and lattice defects, among other features. Site-resolved detection of the populations in this momentum-space lattice is enabled through time of flight absorption imaging.

**Single-site injection.** One method for probing topological bound states of the SSH model is to abruptly expose our condensate atoms, initially localized at only a single lattice site, to the Hamiltonian (equation (1)) and observe the ensuing quench dynamics. When population is injected onto a defect site, we expect to find a large overlap with the mid-gap state, resulting in a relative lack of dynamics as compared to injection at any other lattice site. Our observations using this quench technique are summarized in Fig. 2. In these experiments, population is injected at a single lattice site of our choosing, and the subsequent dynamics are observed with single-site resolution and 10 μs ($\sim 0.01$ $h/t$) time sampling.

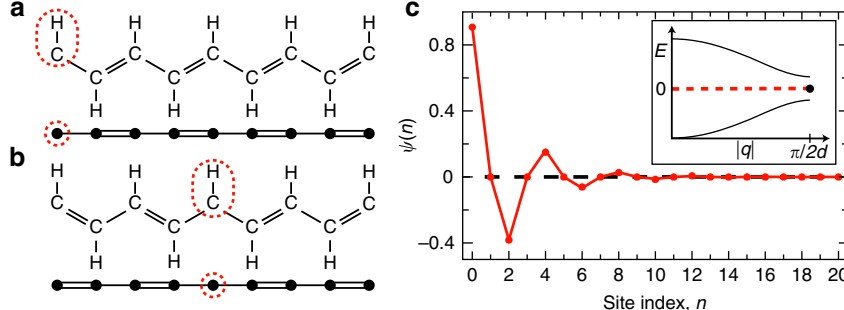

**Figure 1 | The SSH model. (a)** Top: Chemical structure of *trans*-polyacetylene showing the two-site unit cell structure. The dashed red oval encloses the (edge) defect carbon atom at the left system boundary. Bottom: 1D lattice representation of this molecule. **(b)** The two possible topological phases of polyacetylene joined by a (central) defect at the dashed red oval, with lattice representation below. **(c)** Wavefunction of the localized, zero-energy eigenstate versus lattice site index, for the edge defect as in (a) and $\Delta/t = 0.41$. The wavefunction shows population at only even lattice sites (given a defect at site zero) and a $\pi$ phase inversion (reflected in the sign of the wavefunction) at every other even indexed lattice site. Inset: Gapped energy dispersion of the SSH model (equation (1)) for open boundary conditions and a lattice spacing $d$, with a mid-gap state at zero energy.

Figure 2a shows the full dynamics for population injected at the edge defect site, for a lattice characterized by $\Delta/t = 0.40(1)$. We observe slow dynamics and significant residual population in the defect site at long times, suggesting localization at the defect. We additionally observe characteristics of the mid-gap state's parity in these dynamics, that is, the odd lattice sites remain sparsely populated as some atoms spread away from the edge to the second lattice site. Whereas edge injection results in localization, injecting population into the bulk (site five) leads to faster dynamics and increased population spread, as shown in Fig. 2b for $\Delta/t = 0.40(1)$. For these cases of edge and bulk injection, the normalized population dynamics of three sites near the injection point are shown in Fig. 2c,d. The dashed lines represent numerical simulations of equation (1) with no free parameters, exhibiting excellent agreement with the data.

**Phase-sensitive injection.** A more sophisticated probe of the topologically protected mid-gap state can be achieved through controlled engineering of the atomic population prior to quenching the SSH Hamiltonian. Specifically, we can initialize the atoms to match the defining characteristics of a mid-gap state localized to an edge defect: decay of amplitude into the bulk, absence of population on odd lattice sites, and $\pi$ phase inversions on successive even sites. We expect that such an initialization should more closely approximate the mid-gap eigenstate, resulting in the near absence of dynamics following the Hamiltonian quench. However, if the relative phases of the condensate wavefunction at different lattice sites are inconsistent with those of the mid-gap state, significant dynamics should ensue.

We prepare an approximation to the mid-gap state through a two-stage process, as illustrated in Fig. 3a. In the first stage of the sequence, only sites zero and one are coupled, with roughly 35% of the atomic population transferred to site one with a natural phase shift of $\pi/2$, that is, $H_{(1)} = -t(c_1^\dagger c_0 + \text{h.c.})$. Then, in the second stage, sites one and two are coupled to allow full transfer of the population at site one to site two with a chosen phase shift. This second stage is characterized by the Hamiltonian $H_{(2)} = (te^{-\varphi}c_2^\dagger c_1 + \text{h.c.})$, such that the total relative phase between sites zero and two is $\varphi$. Thus, this initialization sequence results in appreciable population at lattice sites zero ($\sim 65\%$) and two ($\sim 35\%$) with a chosen phase difference of $\varphi$ between them.

Figure 3b summarizes the results of our probing the inherent sensitivity of the mid-gap state to this controlled relative phase $\varphi$.

Here, the initial state is prepared with some chosen $\varphi$ and subjected to a quench of the Hamiltonian with $\Delta/t = 0.36(1)$ for an evolution time of $\sim 0.78\ h/t$. When the phase difference of the initial state agrees with that of the mid-gap state ($\varphi = \pm \pi$), the average distance from the edge of the system is minimized. Conversely, a phase difference of zero results in population spreading furthest from the defect site. We see excellent agreement between the full dependence on $\varphi$ and numerical simulations with zero free parameters in Fig. 3b. For the two extremal initialization conditions of $\varphi = \pi$ and 0, example time of flight images and full quench dynamics are depicted in Fig. 3c,e. The quench dynamics shown in Fig. 3d,e more fully illustrate and contrast these two cases. A near absence of dynamics is seen when the phase matches that of the mid-gap state ($\varphi = \pi$), while defect-site population is immediately reduced when the phase does not match ($\varphi = 0$).

**Adiabatic preparation.** Lastly, using our full time-dependent control over the system parameters, we directly probe the mid-gap state through a quantum annealing procedure. We begin by exactly preparing the edge mid-gap eigenstate in the fully dimerized limit of the SSH model, that is, with only the odd tunnelling links present at a strength $t_{\text{odd}} = t + \Delta_{\text{final}}$. Atomic population is injected at the decoupled zeroth site, identically overlapped with the mid-gap state in this limit. Next, we slowly (over a time $\tau_{\text{ramp}} = 1\,\text{ms}$) increase tunnelling on the even links from zero to $t - \Delta_{\text{final}}$, as depicted by the smooth ramp in Fig. 4a and described by the time-dependent Hamiltonian

$$H(\tau) = -\sum_{n \in \text{odd}} (t + \Delta_{\text{final}})\left(c_{n+1}^\dagger c_n + \text{h.c.}\right) \\ - \sum_{n \in \text{even}} t_{\text{even}}(\tau)\left(c_{n+1}^\dagger c_n + \text{h.c.}\right), \tag{2}$$

where $\tau$ denotes time. For adiabatic ramping, the atomic wavefunctions should follow the eigenstate of $H(\tau)$ from purely localized at time zero to the mid-gap wavefunction (for variable $\Delta_{\text{final}}/t$) at the end of the ramp. In the dimerized limit, $\Delta_{\text{final}}/t = 1$, the mid-gap state is isolated from two flat energy bands by a gap energy equal to $t$. This energy gap is reduced as $\Delta_{\text{final}}/t$ decreases, with dispersions as in Fig. 1(c, inset) for intermediate values. This gap finally closes and a single dispersive band emerges as $\Delta_{\text{final}}/t \to 0$.

Figure 4 summarizes our results using this adiabatic preparation method. Both simulated and averaged experimental absorption images for an adiabatically loaded lattice with the

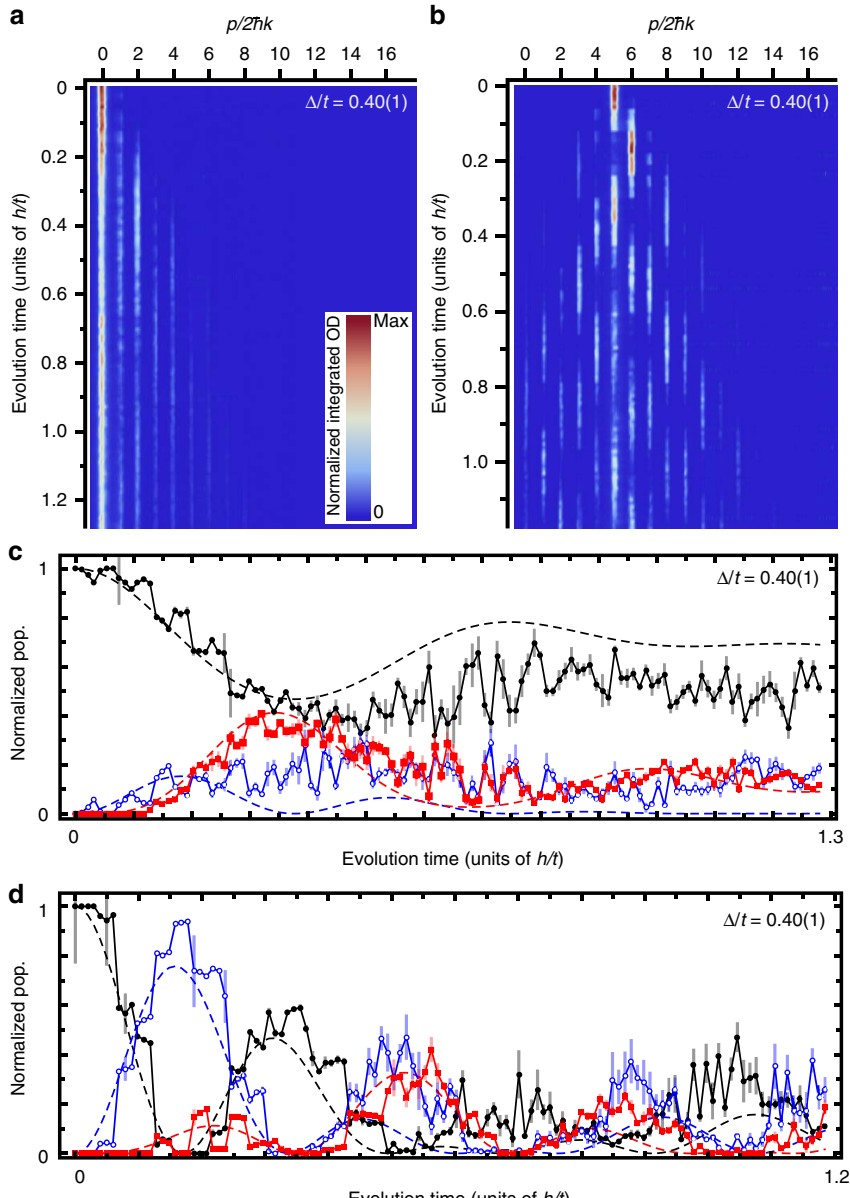

**Figure 2 | Nonequilibrium quench dynamics.** (**a**) Normalized integrated (over the image dimension normal to the momentum-space lattice) optical density (OD) versus evolution time for population injected at the edge defect and $\Delta/t = 0.40(1)$. (**b**) Normalized integrated OD versus evolution time for population injected in the bulk (lattice site five) and $\Delta/t = 0.40(1)$. (**c**) Population versus evolution time for lattice sites zero (black circles), one (red squares) and two (open blue circles) following population injection at the edge defect. (**d**) Population versus evolution time for lattice sites five (black circles), six (red squares) and seven (open blue circles) following population injection in the bulk. Dashed curves in (**c,d**) are solutions to the effective dynamics described by equation (1). All error bars denote one s.e. of the mean.

defect on the left edge are shown in Fig. 4b, demonstrating excellent agreement. Adiabatic preparation was also performed for a lattice with a defect at its centre, and Fig. 4c presents simulated and averaged experimental absorption images for this case, also showing good agreement.

As mentioned earlier and shown in Fig. 1c, the amplitude of the mid-gap state wavefunction is largest at the defect site and decays exponentially into the bulk, owing to the energy gap $\Delta$. In units of the spacing $d$ between lattice sites, the decay length $\xi$ should scale roughly as the inverse of this energy gap (normalized to the average tunnelling bandwidth $t$). We thus expect highly localized mid-gap states for $\Delta/t \sim 1$ (dimerized limit) and an approach to full delocalization over all 21 sites for $\Delta/t \ll 1$ (uniform limit). Using our ability to tune the normalized tunnelling imbalance, we present in Fig. 4d

a direct exploration of the mid-gap state's localization decay length as a function of $\Delta_{\text{final}}/t$. Here, we determine the decay length by fitting the measured atomic populations on even sites at the end of the ramp to an exponential decay. For very small $\Delta_{\text{final}}/t$, we expect the observed decay length to differ from that of the true mid-gap state due to deviations of our ramping protocol from adiabaticity with respect to a vanishing energy gap. Specifically, our smooth ramps of $\Delta_{\text{final}}/t$ should have a duration that greatly exceeds the time scale associated with the smallest energy gap (that is, $\tau_{\text{ramp}} \gg \hbar/\Delta_{\text{final}}$) to remain fully adiabatic. However, our ramp duration is actually shorter than $\hbar/\Delta_{\text{final}}$ for the cases when $\Delta_{\text{final}}/t < 0.13$, and we should thus expect significant deviation from the predictions for the exact (adiabatic) mid-gap state as we approach small values of $\Delta_{\text{final}}/t$. Still, the data in Fig. 4d are in good qualitative

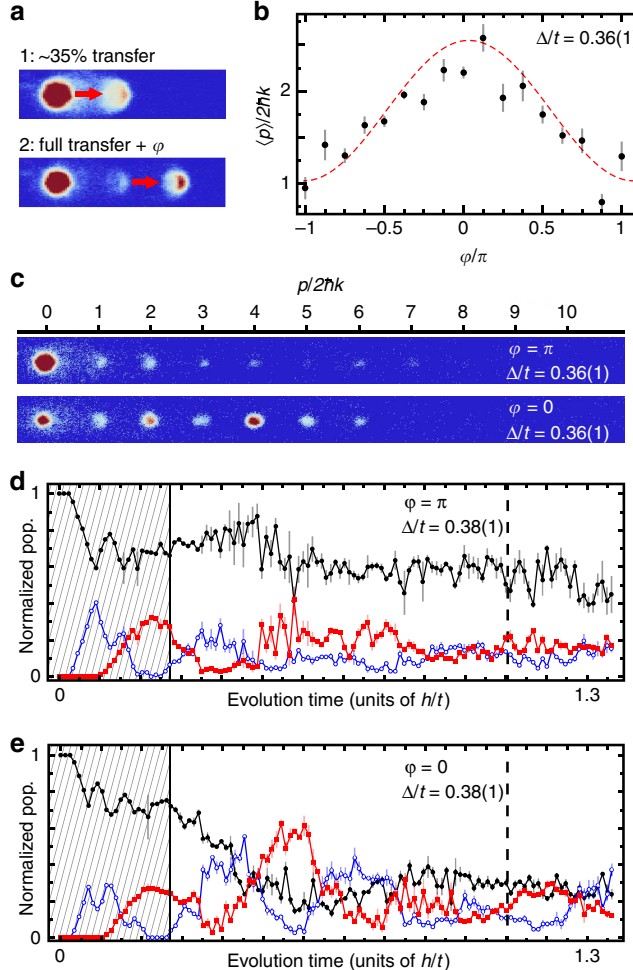

**Figure 3 | Phase-sensitive injection.** (**a**) Absorption images detailing the two-stage state initialization sequence. In stage 1, ~35% of the atoms are transferred from site zero to site one (indicated by the red arrow) with no applied phase shift. In stage 2, nearly all of the atoms in site one are transferred to site two with a controlled phase shift $\varphi$. (**b**) The expectation value of the site index $n$ (average distance from the system's edge) is plotted versus the phase $\varphi$ of initialized states, following a Hamiltonian quench and 760 μs (~0.78 $h/t$) of evolution for $\Delta/t = 0.36(1)$. The dashed line corresponds to a numerical solution of equation (1) given the prepared initial state with no free parameters. (**c**) Absorption images taken after 760 μs (~0.78 $h/t$) of evolution following the initialization and quench, corresponding to phases of $\varphi = \pi$ (top) and $\varphi = 0$ (bottom), respectively, for $\Delta/t = 0.36(1)$. (**d**,**e**) Normalized population at lattice sites zero (black circles), one (red squares) and two (open blue circles) versus quench evolution time for $\Delta/t = 0.38(1)$. The shaded regions and dashed lines denote initialization and imaging stages of the experiment, respectively. All error bars denote one s.e. of the mean.

agreement with the simple expectation of an inverse dependence on $\Delta_{final}/t$, and are mostly consistent with both a numerical simulation of the actual experimental ramping protocol (blue dashed line) as well as predictions based on the exact mid-gap state (red line).

## Discussion

Having observed clear evidence for the topological mid-gap state of the SSH model in the non-interacting limit, we will extend our work to study the stability of this state under the influence

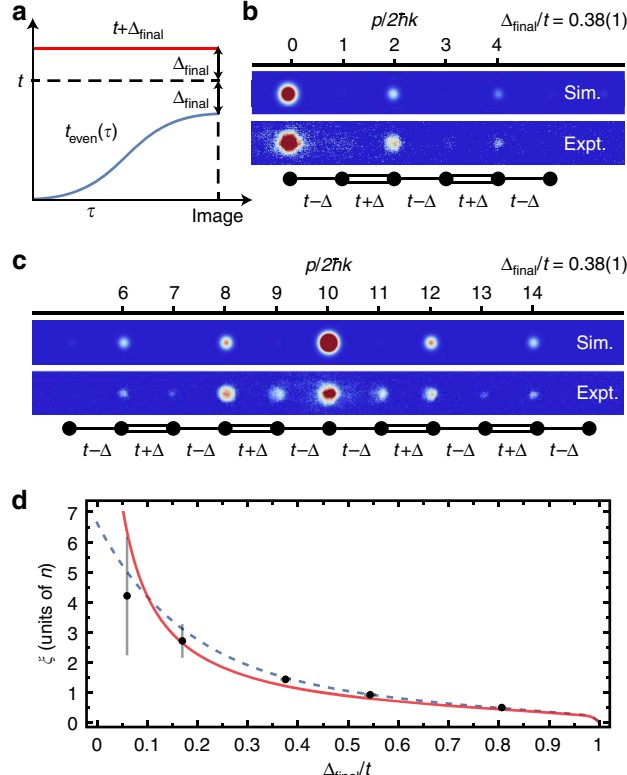

**Figure 4 | Adiabatic preparation.** (**a**) Time sequence of the smooth, 1 ms-long ramp of the weak tunnelling links (blue), holding the strong (red) links fixed with $\Delta/t = 0.38(1)$. (**b**) Simulated (top) and averaged experimental (bottom) absorption images for an adiabatically loaded edge-defect lattice. (**c**) Same as (**b**), but for an adiabatically loaded central-defect lattice. (**d**) Decay length of the atomic distribution on even sites of the edge-defect lattice versus $\Delta_{final}/t$. The dashed blue line represents the results of a numerical simulation of the experimental ramping protocol and the red line shows the exact mid-gap state decay length as a function of $\Delta/t$ for a 21-site lattice. All error bars denote one s.e. of the mean.

of nonlinear atomic interactions. Repulsive, long-ranged (in momentum space) interactions are naturally present in our system due to the atoms' short-ranged interactions in real space, however the present investigation employs large tunnelling bandwidths that dominate over the interaction energy scales. Future explorations of interacting topological wires may be enabled by reducing the imposed tunnelling amplitudes, enhancing the atomic interactions (or their variation in momentum space[27]), or through related techniques based on trapped spatial eigenstates[28] instead of free momentum states[25,26].

In addition, our arbitrary control over the simulated model naturally permits investigations of critical behaviour and quantum phase transitions in disordered topological wires[29]. Topological phase transitions may also be explored in the context of coupled topological wires[30,31] upon extension of our technique to higher dimensions. More generally, given our direct control of tunnelling phases, momentum-space lattices in higher dimensions will allow for the creation of arbitrary and inhomogeneous flux lattices for cold atoms (this control has recently been realised and will be reported elsewhere[32]).

## Methods

**Constructing the momentum space lattice.** Our experiments begin with the creation of $^{87}$Rb Bose-Einstein condensates containing ~$5 \times 10^4$ atoms via

all-optical evaporative cooling, as described in ref. 26. Through time-dependent driving with an optical lattice potential, we initiate controlled momentum-space population dynamics of the condensate atoms amongst 21 chosen discrete plane-wave momentum states. As described in refs 25,26, the 'momentum-space lattice' represented by these 21 states is engineered through the parallel driving of 20 different two-photon stimulated Bragg diffraction processes[33]. For each of these Bragg processes, one of the two relevant interfering laser fields is provided by one of the laser fields composing our optical dipole trap. A counterpropagating laser field is derived from this trap beam, in a manner that allows us to imprint an arbitrary frequency spectrum relative to the forward-propagating beam, as described in ref. 26. By imprinting multiple (20) discrete frequency components onto this counterpropagating beam, with controlled amplitude, frequency, and phase, we are able to simultaneously address the chosen Bragg resonances and implement an effective tight-binding Hamiltonian with full control over all site energies and inter-site tunnelling terms.

**Calibration of tunnelling strengths.** For the presented experiments, we independently measure the tunnelling strength of the strong $(t + \Delta)$ and weak $(t - \Delta)$ links through two-site Rabi oscillations as described in ref. 26. From these values and their standard errors we extract all other reported parameters.

**Imaging.** Since the lattice sites we consider are momentum states of the condensate, they naturally separate along the lattice dimension during time of flight, that is, when all confining potentials have been turned off, allowing for single-site resolution of the atomic populations. All of the data presented herein are extracted from absorption images of the atomic density after 18 ms time of flight, with details of the image and data analysis described in the supplemental material of ref. 26.

**Data availability.** All datasets generated during the performance of this study are available from the corresponding author upon request.

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

## Acknowledgements

We thank T.L. Hughes, I. Mondragon-Shem and S. Vishveshwara for helpful discussions.

## Author contributions

E.J.M. and F.A.A. performed the experiments. E.J.M. analysed the data. All authors contributed to the preparation of the manuscript. B.G. supervised the project.

## Additional information

**Competing financial interests:** The authors declare no competing financial interests.

**Publisher's note**: 

