## [Peer Review File · Nature Communications]

Reviewer #1 (Remarks to the Author):

This paper describes an innovative exercise in engineering quantum Hamiltonians. The authors couple discrete atomic momentum-states so that the momentum-space dynamics mimics the real-space behavior of a particle hopping on a lattice. In particular, they engineer a one dimensional dimerized lattice. This is an influential model, and there is a localized quantum mode at the edge of such a chain. The authors probe the system through a "quantum walk" experiment: They injecting wave-packets at discrete momenta, and watching them evolve. Wave packets injected in the bulk spread. Wavepackets injected at the edge (or at a particular sort of defect) remain localized. They also give a detailed study of a more sophisticated injection scheme.

I have to admit that I have mixed feelings about studies like this. The model they implement is extremely influential, and it is exciting to see an experimental realization. The reason the model is influential however, is that it is very simple, allowing complete theoretical understanding. Modeling this experiment is an undergraduate exercise. The entire challenge lies in getting a very complicated atomic system to behave in such a simple manner.

In the end, I think there is value in such studies, and that it is worthwhile to make concrete the abstract models which guide our thinking. Thus I recommend publication of this paper in Nature Communications.

The actual experiment is well described. I can't say that figure 4(b) or 4(c) really look like "good agreement" between theory and experiment, but other than that the analysis looks sound.

Reviewer #2 (Remarks to the Author):

Meier, An, and Gadway performed a very interesting experiment on creating the Su-Schrieffer-Heeger model in the momentum space and observing the topological soliton. This work is innovative. I support publishing this manuscript in Nature Communications.

Su-Schrieffer-Heeger model has been known as a textbook example in topological matters. The recent realisation of such model in the real space using ultracold atoms provides physicists new opportunities to study such model, which are not available in solids. Whereas the previous scheme in the literature, which made use of a double-well optical lattice, has led to important breakthroughs, like the direct measurement of the Zak phase for the first time in laboratories, here, the authors used a different approach, an atom-optics realization of lattice tight-binding models in the momentum space, based on their previous works (references [25,26]). Such scheme has a number of unique advantages. The single-site (in unit of $4\pi/\lambda$) dressing and detection are very natural. One does not need to make much effort to increase the resolution at all, unlike the real space imaging in optical lattices. The tunneling and its phase could also be easily manipulated. A sharp edge, or interface, can be made straightforwardly.

Using this elegant experimental scheme, the authors have studied the dynamics, phase-sensitive injection, and the adiabatic preparation of the soliton state. To my best knowledge, neither of these had been done in the community of ultracold atoms. I therefore view this work as a very important progress. Moreover, as the authors have clearly explained, their technique may lead to more potential applications and the possibilities of discovering new physics in the near future, ranging from effects of the long-range interactions to coupled wires in higher dimensions. It is convincing that this work shall inspire both theoretical and experimental interests to investigate a wide range of new phenomena that may arise from this system created by the authors. I expect that this work will have considerable impact on both the cold atom and condensed matter community.

Nevertheless, I have a few questions.

1. In figure 2(c) and (d), the agreement between experiment and the simulation based on Eq.(1) at early time looks good. Why the comparison becomes worse in a later time? Also, what is the source for the short time-scale oscillations on top of the smooth profile in experimental data?

2. What is the reason that there are still visible occupations in odd sites, in particular, site 9 and 11, in the adiabatic preparation (see figure 4c, the lower panel for experimental results)?

A rather minor question. What is the colour scale of integrated OD in figure 2(a) and(b)?

I hope that the authors may clarify the above questions to further improve the quality of this nice manuscript.

Reviewer #3 (Remarks to the Author):

The manuscript by E.J. Meier et al reports on the observation of the mid-gap state in Su-Schrieffer-Heeger (SSH) model implemented with an atom-optic method. The author used momentum-space lattice to implement the topological SSH model and studied the transport property of the localized mid-gap state for the first time using atomic system.

Engineering the topological properties of physical systems are of high interest not only in the cold atom community, but also in condensed-matter and photonic communities. I agree that this paper adds novel result to these fields.

The momentum-space lattice can be viewed as another version of a synthetic dimension lattice proposed in Ref [14] and first presented by Florence and NIST groups [(B.K. Stuhl et al, and M. Mancini et al, Science 349 (2015)]. Similar to these experiments, which used atomic spin states instead, the authors were able to prepare the initial states within the "lattice sites" with high controllability. Most of the atomic quantum simulation research these days are limited to system with optical lattices. I believe the idea of these "synthetic dimension" lattices have many potential possibilities, and could have high-impact in the field. With the authors' demonstration of the topological lattice model, the authors indeed show the strength of their momentum-space (synthetic) lattice for quantum simulation.

Regarding the readability of the paper, I think it is well written for non-specialist.

I have some questions and comments on the manuscript.

Line 271: The authors claimed that interaction is negligible in the system. Could the authors give energy scale (e.g. U/t) of the (long-range) interaction?

Is there any inhomogeneity in the lattice sites that affects site-dependent tunneling (t) and offsets (Δ)? In a real space optical lattice, the external confinement can potentially create a localized state at the edge due to the energy offset between the nearest sites.

Could the authors use a band picture (e.g. Inset in Fig. 1c) to explain how can the mid-state be prepared by the adiabatic preparation method.

Line 237: How did the authors confirmed their adiabaticity? What is the value of tunneling parameter t here? In this section, the author did not give the time scale in unit of \hbar/t . It seems to me that the 1 ms ramp time could be much shorter than the effective tunneling time.

In Fig. 4, I do see the simulation and the data fit well. However I do not see the $(\Delta / \text{it}\{t\})^{-1}$ scaling in the simulation, nor in the data. Could the authors explain why? If the authors can show a theory curve derived from SSH model, it would be more convincing.

Before I recommend this paper for publication in Nature Communication, I would like the authors to address these questions.

Response to all Reviewers

We thank all Reviewers for their time and effort, and especially for their valuable comments and criticisms regarding our manuscript. In the following pages we individually address the comments of Reviewers 1, 2, and 3, and we hope that these point-by-point responses will convince the Reviewers that our resubmitted manuscript is suitable for publication in *Nature Communications*. Before proceeding to these responses, we briefly enumerate any changes to the manuscript not solicited or motivated by the Reviewers.

Unsolicited Changes

We have added the following reference to our manuscript:

[30] Xiong-Jun Liu, Zheng-Xin Liu, and Meng Cheng, "Manipulating topological edge spins in a one-dimensional optical lattice," *Phys. Rev. Lett.* 110, 076401 (2013).

This referenced work predated and was quite similar to a theory paper cited in our original submission, and so we have included this earlier reference so as not to improperly misattribute credit for theoretical work on the two-leg Su-Schrieffer-Heeger model.

Response to Reviewer 1

We thank the Reviewer for their valuable comments and criticisms related to our submitted manuscript. We are pleased that the reviewer has a favorable view of this work, and is in support of its publication in *Nature Communications*. Below we provide a point-by-point response to the Reviewer's comments, criticisms, and questions, and when relevant we detail any related changes made to the revised manuscript.

Response to comments / changes to the manuscript

Comment: "I can't say that figure 4(b) or 4(c) really look like "good agreement" between theory and experiment, but other than that the analysis looks sound."

Response: We thank the Reviewer for their opinion on our general analysis. We agree that the agreement between data and simulation in figure 4(c) is less than ideal. However, we tend to believe that the agreement between the data and theory in figure 4(b) is actually quite good – the exponential decay of population into the bulk is reproduced with good agreement, and little to no population is observed on sites of the wrong parity.

We believe that our initial presentation of the simulated absorption images, which simply convoluted the simulated populations (determined by numerical simulation of the SSH model according to the experimental tunneling parameter dynamics) with a somewhat arbitrary 2D Gaussian function, did a poor job of representing the good agreement between the measured and simulated *population* distribution. In particular, the size of the Gaussian function used to present the simulated distribution was not well matched to the observed time-of-flight momentum spread of our condensate atoms. We have updated these simulated atomic distributions accordingly, to more faithfully present what we believe is good (not excellent, however) agreement between the experimental data and the simulations (which are free of any fit parameters).

Changes to manuscript: The theoretical contributions to figures 4(b) and 4(c) have been updated.

Response to Reviewer 2

We thank the Reviewer for their valuable comments and criticisms related to our submitted manuscript. We are pleased that the Reviewer has a favorable view of this work, and is in support of its publication in *Nature Communications*, provided that we make the appropriate modifications to address the Reviewer's comments. Below we provide a point-by-point response to the Reviewer's comments, criticisms, and questions, and when relevant we detail any related changes made to the revised manuscript.

Response to comments / changes to the manuscript

Comment: "In figure 2(c) and 2(d), the agreement between experiment and the simulation based on Eq. (1) at early time looks good. Why the comparison becomes worse in a later time? Also, what is the source for the short time-scale oscillations on top of the smooth profile in experimental data?"

Response: In regard to the first question, as our atoms spend more time evolving under the simulated Hamiltonian, a number of effects lead the experimental dynamics to deviate from the theoretical predictions. We believe the most dominant and relevant of these effects to be separation of the atomic wave packets that have different momenta. The coherent, laser-driven dynamics of population between these momentum orders relies on spatial overlap of the their respective real-space distributions. Because the momentum orders by construction have different momenta, they fly apart from each other in real space during the course of the experiment. The relevant timescale for this loss of spatial overlap, i.e. the time when we depart from near-field dynamics, is determined by the coherence length of our condensate atoms (mainly dictated by the condensate extent in 3D at these temperatures) and by their spread of velocities. In these simulations of one-dimensional systems, for the tunneling rates (two-photon Bragg transition rates) that we employ, and for the experimental details of our condensate, we expect to observe at least partially coherent dynamics up until approximately 10-20 tunneling times. The simulations, we allow for no free parameters, can also deviate from the data at long times due to errors in the experimentally-calibrated tunneling rates.

We now address the Reviewer's second question, regarding the short time-scale oscillations in the data. We effectively realize a 21-site tight-binding Hamiltonian by separately, resonantly addressing 20 individual Bragg transitions. The exact realization of the tight-binding Hamiltonian is approached in the limit that the tunneling rates are much smaller in magnitude than the actual frequency spacing between adjacent Bragg resonances. If we are not in the weak-coupling limit, the 19 additional off-resonant fields can have a residual influence on the dynamics of a given transition. In particular, for the linear spacing between our Bragg transition resonances (determined essentially by the first-derivative of the quadratic free-particle dispersion), these off-resonant contributions manifest as step-like variations of the actual populations. The time between step-like jumps of the populations is completely determined by the frequency spacing between adjacent Bragg resonances (i.e. by the lattice wavelength and the mass of the atom). However, the magnitude of population transferred during of each step and the ratio between the tunneling time and the "step-time" is determined by the strength of the Bragg coupling (i.e. the tunneling rate). As the tunneling rate becomes smaller, these step-like variation, while still present, become a less distinct aspect of the population dynamics. For our current experimental parameters, however, in light of the aforementioned considerations of separation of momentum orders, we choose to operate with not completely ignorable step-like variations.

For context, we include below a few figures that highlight the step-like population variations that can occur. These data are from a different study on continuous-time quantum random walks (QRW) on a lattice with homogeneous tunneling strengths. We note, however, that these data represent a case with slightly larger tunneling rates than in the submitted manuscript.

The figure just below shows the population in the zeroth momentum order as a function of time (in normalized units) in the case of a QRW.

Finally, the two additional figures just below show the full momentum distribution dynamics of the same QRW on this a 21-site lattice, as well as the measured standard deviation (width) of the momentum distribution (which, neglecting the steps, increases linearly with time as expected for a QRW). Here, coherent step-like behavior can be seen in global observables almost up until the atoms reach the edge of the lattice (when the atoms reflect and the momentum-spread begins to decrease).

Changes to manuscript: N/A

Comment: “What is the reason that there are still visible occupations in odd sites, in particular, site 9 and 11, in the adiabatic preparation (see figure 4c, the lower panel for experimental results)?”

Response: Although the ramp is certainly slow enough to see qualitative agreement with the mid-gap state, the aforementioned decoherence (due to momentum-order separation) can limit our ability to go as slow as necessary to achieve perfect adiabaticity. For large values of Δ/t our 1 ms ramp is roughly 3-4 times the time scale associated with the smallest energy gap (Δ_{final}), allowing us some degree of adiabaticity. However, for small values of Δ/t the ramp would have to be made much longer to keep this same ratio of ramp time to energy gap. Since our momentum states are separating in space during the course of the experiment, eventually we reach a point where they are no longer overlapped enough in space to allow for coherent evolution. This decoherence puts a hard time limit on our experiment, such that we are not able to make the ramp arbitrarily long and perfectly adiabatic so as to assure complete agreement with the theoretical simulations.

Changes to manuscript: N/A

Comment: “What is the colour scale of integrated OD in figure 2(a) and 2(b)?”

Response: The data here is actually not just integrated the integrated optical depth (OD), but it is also normalized to the total atom number within each image. Because the present studies are mainly interested in studies of the single-particle dynamics of the SSH model, and thus the actual atom number during any individual run is not particularly germane, this helps to avoid fluctuations to presented data due to irrelevant shot-to-shot variations in the atom number (which, however, are not particularly large). Explicitly, the one-dimensional distribution of integrated OD values is normalized at the pixel level so that the sum over pixel values equals one.

Furthermore, for a full color plot that includes many different evolution times, the color scale is set such that the maximum value matches to the maximum normalized integrated OD value found at any time. As the population is initially localized to a single momentum-order, this maximum color value tends to be found at the shortest times, and the population tends to spread out as time proceeds. So while the actual maximum value of the color scale is related to the measured ODs, the numerical value itself is not particularly important, as it depends on, for example, the pixel size and magnification of our imaging system.

Changes to manuscript: To reflect these subtleties, the color scale (which is representative of the color scale of all figures in the paper) is changed to be labeled as “Normalized Integrated OD” and the labels “0” and “Max” are written next to the respective low and high levels of the color scale. We thank the Reviewer for pointing out the ambiguity of our earlier formulation of this color scale.

Response to Reviewer 3

We thank the Reviewer for their valuable comments and criticisms related to our submitted manuscript. We are pleased that the Reviewer has a favorable view of this work, and has given us the opportunity to further clarify some points. Below we provide a point-by-point response to the Reviewer's comments, criticisms, and questions, and when relevant we detail any related changes made to the revised manuscript. We hope that these responses will convince the Reviewer of the merit of our manuscript and that they will be in favor of its publication in *Nature Communications*.

Response to comments / changes to the manuscript

Comment: "Line 271: The authors claimed that interaction is negligible in the system. Could the authors give energy scale (e.g. U/t) of the (long-range) interaction?"

Response: For our atom numbers and trapping frequencies, we can estimate the chemical potential energy associated with our condensate to be $\mu/\hbar = 2\pi \times 660$ Hz [based on Dalfovo et al. RMP 71, 463 (1999)]. This chemical potential energy, representing the maximum mean-field interaction energy in the highest density region of our condensate, ends up being about one-half of our typical tunneling energy, $t/\hbar \sim 2\pi \times 1.3$ kHz. While at first blush this seems like it should lead to a significant influence of atomic interactions, one has to consider that in the relevant "synthetic lattice" composed of momentum states the local (in real-space) atom-atom interaction due to s -wave scattering should be extremely long-ranged. If one considers a δ -like interaction, it would give rise to an infinite-ranged in momentum-space. In the limit of an infinite-range interaction, all possible population distributions amongst the available momentum states will have exactly the same interaction energy. Thus, in this limit, interactions should play no role on the dynamical response of the system.

However, given the large spread of relevant momentum states that may be populated in our system, one actually expects a nontrivial variation of the effective scattering properties due to the large center-of-mass collision energy of atoms in different momentum orders. For atoms with relative momentum $\hbar k_{rel}$, the scattering cross-section will be given by: $\sigma(k_{rel}) = \sigma_0 / (1 + (k_{rel}a)^2)$, where a is the scattering length for rubidium-87 and $\sigma_0 = 8\pi a^2$ is the low-energy limit of the s -wave scattering cross-section. We can relate k_{rel} to the distance of atoms in the "lattice" (in units of the site-index n) as $k_{rel}(n) = 2nk$, with k the wavevector associated with our Bragg laser light. Below, considering this dependence, we plot the effective variation of the scattering cross-section as a function of the distance between atoms in units of the lattice site index, based on our current experimental configuration with $\lambda = 1064$ nm and with ^{87}Rb .

For the given system sizes and experimental parameters, this variation is quite long-ranged, as is the associated variation of the interaction energy between different momentum modes. Some straightforward ways to be sensitive to these interaction energies, however, would involve looking over larger system sizes and longer evolution times, performing investigations with lower tunneling rates, by resonantly enhanced interactions through a Feshbach resonance, or to actually exploit resonant nonlinear processes (such as four-wave mixing) as a different type of interaction-driven influence that would not be naturally long-ranged.

Changes to manuscript: N/A

Comment: “Is there any inhomogeneity in the lattice sites that affects site-dependent tunneling (t) and offsets (Δ)? In a real space optical lattice, the external confinement can potentially create a localized state at the edge due to the energy offset between the nearest sites.”

Response: In our realization of the effective tight-binding model, based upon the resonant addressing of many individual Bragg transitions, there is no effective inhomogeneity of the lattice site energies (although we can directly engineer such an inhomogeneity, as demonstrated in our earlier study through the observations of Bloch oscillations in a “tilted” lattice). In particular, every Bragg transition is driven perfectly on resonance, and the influence of the small off-resonant terms is negligible, as they also mainly cancel due to the linear spacing of the addressed Bragg resonances.

Regarding any real-space potentials, such as the optical trapping potential and any inhomogeneities of the Bragg laser beams, they essentially have no effect on the matter-wave dynamics at the short times we consider. At much longer times, of order the trap period, the remaining optical potentials can have the effect of causing atoms with non-zero momentum to decelerate as they move up the trapping potential, thus dispersing them to momentum states (no longer taking only discrete values) that are not directly addressed by the applied Bragg resonance transitions.

Changes to manuscript: N/A

Comment: “Could the authors use a band picture (e.g. Inset of Fig. 1c) to explain how can the mid-state be prepared by the adiabatic preparation method.”

Response: As suggested by the Referee, we are happy to elucidate this adiabatic preparation in terms of the gapped band structure, and we have included a similar line of argumentation in the manuscript text as well.

A series of band structure diagrams is displayed below, on the following page. In the fully dimerized limit, when $\Delta = t$, the bulk bands are completely flat, relating to isolated symmetric and asymmetric phase dimer configurations. For one isolated boundary site, decoupled from all other sites, we find a corresponding mid-gap state that exists at zero energy between these two flat bands. In this limit, the mid-gap state’s spatial wavefunction is trivially localized only to the isolated boundary site.

As Δ/t is decreased, and these dimers are coupled and a continuous lattice is formed, this bandgap begins to close and the spatial wavefunction of the mid-gap state starts to extend into the bulk with an exponential decay. The length scale associated with this decay is essentially set by the ratio of the tunneling energy to the energy gap to the dispersive bulk states. When the limit of homogeneous tunneling is reached, for $\Delta/t = 0$, the bandgap fully closes and the mid-gap state becomes fully dispersive. In this limit, only one dispersive energy band remains, as the length scale associated with the alternating tunneling (twice the lattice spacing) has been removed.

We have the ability to initialize the population at the system boundary when the mid-gap state is trivially localized to this location ($\Delta/t = 1$ in the top left of the figure), and by slowly varying the Hamiltonian of the system (with respect to the relevant energy gaps), we can maintain population in this particular energy eigenstate even as its character (i.e. its spatial wavefunction) changes upon a decrease of Δ/t .

Changes to manuscript: A few sentences describing the “quantum annealing”, or adiabatic preparation, procedure in terms of the system’s band structure has been added to the manuscript in paragraph 1 of the Adiabatic Preparation section, just after Eq. (2). We thank the Reviewer for this excellent suggestion.

Comment: “Line 237: How did the authors confirmed their adiabaticity? What is the value of tunneling parameter $\{t\}$ here? In this section, the author did not give the time scale in unit of $h/\{t\}$. It seems to me that the 1 ms ramp time could be much shorter the effective tunneling time.”

Response: We thank the Reviewer for pointing out this deficiency, that we nowhere in the manuscript mentioned the magnitude of the tunneling strength (or tunneling time) in physical units.

As a brief aside, we note that at the time that this data was taken, we had a simple and technical restriction on the duration of our experimental evolution times to around 1 ms, set by the longest (in terms of data string) arbitrary waveforms that we could create, combined with the waveforms' use to generate an arbitrary frequency spectrum (at frequencies near 80 MHz, then written onto the laser with an acousto-optic modulator) necessary to controllably address 20 separate Bragg resonances. We can now perform experimental sequences roughly ten times longer than this.

It turns out that this technically-limited time of 1 ms is in fact not much longer than the typical timescale over which we expect to lose spatial overlap, and thus coherence, of our different momentum orders. So, even if we could have ramped system parameters more slowly in these experiments, we still very likely would have found deviations from the true mid-gap state, due to the loss of coherence as opposed to nonadiabaticity. Future experiments, based on condensates with longer coherence lengths, will be able to faithfully access coherent dynamics over longer timescales.

We now address the Reviewer's main point. A perfectly adiabatic ramp would be based on a ramp duration in great excess of the time scale associated with the system's energy gap. Below, we plot and compare directly the ramp duration τ_{ramp} , in units of $\hbar/\Delta_{\text{final}}$ that are normalized to the relevant minimal energy gap, as a function of Δ_{final}/t . We find that the ramp duration actually becomes lower than the time scale associated with this energy gap for values of $\Delta_{\text{final}}/t < 0.13$ (indicated by the gridlines on the plot below).

Therefore, some of our measurements were certainly not perfectly adiabatic, and we would expect to find a large deviation from the mid-gap state for very small Δ_{final}/t , as mentioned in the submitted manuscript. We now make this discussion of the relevant timescales much more explicit, and we are indebted to the Reviewer for helping to strengthen this discussion about the adiabatic preparation. To note, as addressed in the following comment, we now include a theory curve that shows the predicted decay length of the actual mid-gap state (which would be achieved for a perfectly adiabatic ramp) as well. We find that the data taken, with their experimental error bars, end up being consistent with both the adiabatic and actual, slow but finite-duration ramping procedure.

Changes to manuscript: In the manuscript, we now present in much greater detail a discussion about the limitations of our 1 ms ramp, with respect to adiabaticity and faithful preparation of the mid-gap state for different Δ_{final}/t values.

Comment: “In Fig. 4, I do see the simulation and the data fit well. However I do not see the $(\Delta/t)^{-1}$ scaling in the simulation, nor in the data. Could the authors explain why? If the authors can show a theory curve derived from SSH model, it would be more convincing.”

Response: The main deviation of data and simulation from an inverse dependence is at low values of Δ/t . Here, we attribute the failure of the simulation to reach such high values to the reduction in adiabaticity due to the vanishing energy gap. We have produced another theory curve derived from the decay length of the true midgap state of the SSH Hamiltonian, i.e. that which would be reached by a perfectly adiabatic ramping procedure and fully coherent dynamics. This new theory line, along with the data and the prediction based on the actual experimental ramping protocol are shown below.

The dashed blue line in the figure represents the original simulation of the SSH model under the slow ramping protocol. The new red line represents the decay length of the SSH mid-gap. As can be seen from this new plot, the data are generally consistent with either curves, and expected influence of nonadiabaticity of the slow ramping protocol can be deduced from the difference in the two theory curves.

Changes to manuscript: The updated graph shown above is now included in the manuscript with the additional theory curve. We thank the Reviewer for leading us to include this extremely relevant comparison curve.

Reviewer #1 (Remarks to the Author):

The authors addressed all of my concerns, and I recommend publication.

Reviewer #2 (Remarks to the Author):

I have read the author's reply and the revised manuscript. All the answers to my questions seem reasonable. It is understandable that a finite ramping rate and finite off-resonant transitions contribute to the quantitative disagreement between experiments and theory in certain time scales. Nevertheless, qualitative results that are relevant to the underlying physics are rather clear. As I stated in my first report, this experimental work is interesting and innovative. I do recommend publishing it in Nature Communications.

I note that the authors mentioned that they had added reference [30] in the cover letter. The reason is that "since it is predated and was quite similar to a theory paper cited in our original submission, and so we have included this earlier reference so as not to improperly misattribute credit for theoretical work on the two-leg Su-Schrieffer-Heeger model." In the manuscript, they actually added another reference [31] together with [30]. Being interested at knowing what references that the authors previously missed, I went through references [30] and [31]. However, I did not find out any discussions in these two references on two-leg Su-Schrieffer-Heeger model. Since citing references incorrectly will be very misleading for readers who may not check the original references carefully, I strongly suggest that the authors should carefully examine the relevance of the cited works, and find out a proper way to cite if they want to add references.

Reviewer #3 (Remarks to the Author):

The authors have carefully addressed all my comments, provided detailed answers to my questions, and revise the manuscript property. Especially, in the adiabatic preparation section, the band structure picture they gave and the revised figure in Fig. 4b will definitely help the reader understand the procedure and limitation on the adiabatic loading.

As I have already mentioned in my first review, I think the paper is would be of interest not only for the experts in the field, but also for readers in wider range of areas. Therefore, I strongly recommend the manuscript for publication in Nature Communication.

Response to all Reviewers

We thank all Reviewers for their time and effort, and especially for their valuable comments and criticisms regarding our manuscript. In the following pages we individually reply and send final comments to Reviewers 1, 2, and 3. When necessary, we briefly enumerate any changes to the manuscript in response to the Reviewer comments.

Response to Reviewer 1

We thank the Reviewer again for their time and effort spent in reviewing our manuscript, and we thank them for all of their valuable comments and criticisms.

Reviewer 1 Comment - "The authors addressed all of my concerns, and I recommend publication."

Response to Reviewer 2

We thank the Reviewer for the time and effort spent on reviewing our manuscript, and for their helpful comments and criticisms. We are very thankful that the Reviewer recommends publication in *Nature Communications*.

We respond below to the specific last criticism made by the Reviewer regarding two citations made in the manuscript.

Reviewer 2 Comment - “I note that the authors mentioned that they had added reference [30] in the cover letter. The reason is that “since it is predated and was quite similar to a theory paper cited in our original submission, and so we have included this earlier reference so as not to improperly misattribute credit for theoretical work on the two-leg Su-Schrieffer-Heeger model.” In the manuscript, they actually added another reference [31] together with [30]. Being interested at knowing what references that the authors previously missed, I went through references [30] and [31]. However, I did not find out any discussions in these two references on two-leg Su-Schrieffer-Heeger model. Since citing references incorrectly will be very misleading for readers who may not check the original references carefully, I strongly suggest that the authors should carefully examine the relevance of the cited works, and find out a proper way to cite if they want to add references.”

Response to comment

Response: First, we strongly apologize for having added two new citations when our “statement of unsolicited changes” only detailed one such addition. This misstatement resulted from an unfortunately poor communication on this point between the persons preparing the manuscript (Meier) and the “Reviewer Reply / List of Changes” (Gadway) documents. We did not mean to sneak in an extra citation, but rather intended to cite one of the two works (with one common author who had contacted us), which in fact concerned the same subject. We have removed the reference [31], because of the fact that it was indeed somewhat superfluous, given our addition of reference [30].

However, reference [30] is relevant to the physics of the coupled two-leg Su-Schrieffer-Heeger model, and we believe that it’s inclusion and the manner in which it is referenced in our manuscript is not inappropriate. The paper (reference [30]) uses a different set of terminology when describing the physical system, but it in fact realizes almost exactly the same physical situation as described in our current (this submitted version) reference [31]. This is part of the reason why we never came across it in the first place when delving deeper into the physics of the Su-Schrieffer-Heeger model. Both references ([30] and [31] of the current submitted manuscript) describe a system in which two internal spin states are used as effective “legs” of a “two-leg ladder”. Atoms in these two spin states are able to realize topologically distinct one-dimensional wires. In each case, field-driven transitions are used to “couple” these two “ladder legs”, and a transition of the system from topological to normal is expected as a function of the “inter-leg” coupling strength. While the phrase “Su-Schrieffer-Heeger model” is not used once in reference [30], and instead they discuss the system as a chiral symmetrical topological insulator, there is no distinction between these alternate descriptions in the case that the lattice site positions are fixed in space (no lattice vibrations). Thus, these two references probe much the same physics. We show below two key figures from the two references, each of which demonstrates the coupling of internal states, as

well as the modification of the energy spectrum from having two topological “mid-gap”, zero-energy eigenstates to having no topological character. We believe that the similarity of these two figures, and the subject of these two papers, reflect that they are both highly relevant to the discussed prospect of future studies on coupled topological wires.

Figure 1 from Ref.[30] by X.-J. Liu, Z.-X. Liu, and M. Cheng, PRL (2013)

Figure 1 from Ref.[31] by S.-L. Zhang and Q. Zhou, arXiv (2016)

Changes to manuscript: We have removed reference [31] from the previous manuscript version.

Response to Reviewer 3

We thank the Reviewer again for their time and effort spent in reviewing our manuscript. We are very happy to see that the changes we have made, in response to their helpful comments, have helped us to make a strong case for publication in *Nature Communications*. We thank them again very much for their valuable comments and criticisms.

Reviewer 3 Comment - “The authors have carefully addressed all my comments, provided detailed answers to my questions, and revise the manuscript property. Especially, in the adiabatic preparation section, the band structure picture they gave and the revised figure in Fig. 4b will definitely help the reader understand the procedure and limitation on the adiabatic loading.

As I have already mentioned in my first review, I think the paper is would be of interest not only for the experts in the field, but also for readers in wider range of areas. Therefore, I strongly recommend the manuscript for publication in Nature Communication.”